

# Screening of immunosuppressive factors for biomarkers of breast cancer malignancy phenotypes and subtype-specific targeted therapy

Ping Wang[1,*], Jiaxuan Liu[1,*], Yunlei Song[2], Qiang Liu[3], Chao Wang[4], Caiyun Qian[5], Shuhua Zhang[6], Weifeng Zhu[5], Xiaohong Yang[5], Fusheng Wan[5], Zhuoqi Liu[5] and Daya Luo[5,7]

[1] Queen Mary School, Nanchang University, Nanchang, China
[2] Key Laboratory of Prevention and Treatment of Cardiovascular and Cerebrovascular Diseases of Ministry of Education, Gannan Medical University, Ganzhou, China
[3] National Cancer Center/Cancer Hospital, Chinese Academy of Medical Sciences and Peking Union Medical College, Beijing, China
[4] School of Basic Medical Sciences, Nanchang University, Nanchang, China
[5] Department of Biochemistry and Molecular Biology, School of Basic Medical Sciences, Nanchang University, Nanchang, China
[6] Jiangxi Cardiovascular Research Institute, Jiangxi Provincial People's Hospital, Nanchang, China
[7] Jiangxi Province Key Laboratory of Tumor Pathogens and Molecular Pathology, Nanchang University, Nanchang, China
* These authors contributed equally to this work.

Corresponding authors
Zhuoqi Liu, liuzhuoqi@ncu.edu.cn
Daya Luo, luodaya@ncu.edu.cn

## ABSTRACT

We aimed to screen and validate immunosuppressive factors in luminal- and basal-like breast cancer cell lines and tissue samples associated with malignant phenotypes. The mRNA microarray datasets, GSE40057 and GSE1561, were downloaded and remodeled, and differentially expressed genes were identified. Weighted gene co-expression network analysis (WGCNA) and gene ontology (GO) and KEGG pathway enrichment analysis were performed to explore the immune-related events related to the basal-like breast cancer. The online resources, GOBO, Kaplan–Meier Plotter and UALCAN, were employed to screen for immunosuppressive factors associated with breast cancer malignant phenotypes. Immunohistochemistry was used to evaluate *VEGFA* and *MIF* levels in breast tumors and normal breast tissues; qPCRs and western blots were used to validate the expression of clinical immuno-oncology (IO) therapeutic targets *CD274* (*PD-L1*) and IL8 in cell lines. The results showed that various immune-related events contribute to basal-like breast cancer. First, *TGFβ1* and *IL8* had higher average expression levels in more malignant cell lines; second, *MIF* and *VEGFA* had higher average expression levels in more malignant breast cancer tissues, and the high expression levels were associated with poor survival rate. Third, IO targets *CD274* and *IL8* which were confirmed to be more suitable for the treatment of basal-like breast cancer. In view of the above, during the formation and development of breast cancer, immune-related genes are always activated, and immunosuppressive factors, *IL8*, *TGFβ1*, *MIF*, and *VEGFA* are up-regulated. Such molecules could be used as biomarkers for breast cancer prognosis. However, because individual immune-related factors can play several biological roles, the mechanistic relationship between immunosuppressive factors

and breast cancer malignant phenotypes and the feasibility of their application as drug targets require further investigation.

# INTRODUCTION

Breast cancer, the most frequently diagnosed cancer and the second most fatal cancer in women around the world, affects one in eight women (*American Cancer Society, 2016*). With the advent of gene expression profiling over the last 15 years, breast cancers have been classified into luminal A, luminal B, human epidermal growth factor receptor 2 (HER2 or ERBB2)-enriched, basal-like, and claudin-low categories (*Liu & Wang, 2015*; *Prat et al., 2015*). Among these categories, basal-like breast cancer has garnered significant attention among researchers, as it accounts for ~75% of the highly malignant triple-negative subtype. This biologically aggressive neoplasia takes on several malignant phenotypes, including early onset, high histological grade, increased distant recurrence and visceral metastases, insensitivity to endocrine and targeted therapy, and poor prognosis (*Bahnassy et al., 2015*). Although biomarkers for breast cancer prognosis and therapy (*Jézéquel et al., 2012*) have markedly improved treatment decisions, inconsistent diagnostic criteria for basal-like breast cancer and controversial research findings necessitate the discovery of more specific molecular markers (*Tomao et al., 2015*).

Malignant tumor phenotypes, such as invasiveness, metastasis, drug resistance and poor prognosis, depend on both the distinct genetic and epigenetic characteristics of the tumor as well as other factors in the tumor microenvironment (*Gandellini et al., 2015*). The tumor microenvironment is composed of tumor cells, various types of stromal cells and the extracellular matrix (ECM), in which tumor cells and stromal cells interact by releasing a variety of cytokines, chemokines, and growth factors (*Xu et al., 2012*). In recent years, it has become clear that tumor cells, as well as other cells and factors that accumulate in tumor-bearing hosts, play a critical role in patient outcomes (*Schlößer et al., 2014*). On the one hand, a variety of immune cells can be induced to kill tumor cells. On the other hand, tumor cells have many strategies for escaping immune attack, including the release of immunosuppressive factors. The presence of immunosuppressive factors induces local immune escape in the tumor microenvironment, which thwarts antitumor immune responses and poses a major obstacle to many immunotherapeutic or conventional therapeutic approaches. Fortunately, the high expression levels of these immunosuppressive factors may also be studied to identify therapeutic targets. Currently, multiple immune-oncology (IO) therapeutic targets are available in clinical therapy (*Szekely et al., 2018*). However, due to the multifaceted functions of many immunosuppressive factors in different tumor types and stages of development, there remains controversy regarding their actual and fundamental roles in tumor pathology. This ambiguity

prevents the clinical application of immunosuppressive factors as diagnostic and therapeutic biomarkers.

In this article, by comparing gene expression patterns between basal- and luminal-like breast cancer cell lines and tissue samples, which have different levels of aggressiveness and malignancy, we sought to screen for and verify the immunosuppressive factors associated with a malignant phenotype and investigate the significance of IO targets in the clinical treatment of breast cancer. These immunosuppressive factors could be used as additional markers to identify malignant breast cancer and further tailor therapies for individual breast cancer patients.

## MATERIALS AND METHODS

### Gene expression microarray analysis

The expression monitoring array raw data were downloaded from the gene expression omnibus (GEO) database (Barrett et al., 2012) with accession numbers GSE40057 (Luo et al., 2013) and GSE1561 (Farmer et al., 2005). GSE40057 included 10 breast cancer cell lines and two immortalized breast epithelium cell lines analyzed with the Affymetrix Human Genome U133 Plus 2.0 Array; the GSE1561 data contained 49 breast cancer tissue samples that were analyzed on Affymetrix U133A chips. Principal components analysis (PCA), $K$-means clustering and differentially expressed gene (DEG) screening was performed in R using Bioconductor and associated packages (Gentleman et al., 2004).

### Weighted gene co-expression network analysis and gene ontology and KEGG pathway enrichment analysis

For genome-wide expression profile data of tissues and cell lines, the missing values were first removed, and the genes with average expression levels less than 0.5 were filtered out. Second, all samples performed well in hierarchical clustering, and no outliers needed to be removed. The step-by-step method of the weighted gene co-expression network analysis (WGCNA) package (Langfelder & Horvath, 2008) in R was used to construct the module and co-expression network. Soft thresholds were generated by the pickSoftThreshold function of the WGCNA package, with tissue data set to 28 and cell line data set to 10. The adjacency matrix and the topological overlap matrix (TOM) was calculated according to the corresponding soft threshold. Based on TOM, the corresponding dissimilarities between each gene were calculated, and 400 genes were randomly selected for TOM visualization. In addition, we constructed a hierarchical cluster tree of all genes based on the dissimilarity matrix. Using the dynamic tree cut method, the branches of the hierarchical cluster tree were cut to identify modules. Subsequently, with the hierarchical clustering data of the eigengene module data, a height cutoff of 0.25 was chosen, and similar modules were merged to build the final co-expression network. Finally, the eigengene module data were visualized and module-trait associations were quantified. Gene ontology (GO) and KEGG pathway enrichment analyses were performed for genes in each module using the DAVID functional annotation clustering tool (http://david.abcc.ncifcrf.gov) (Huang, Sherman & Lempicki, 2009).

## GOBO analysis

GOBO (http://co.bmc.lu.se/gobo/), an online resource with mRNA microarray profiling data from 51 breast-derived cell lines (Ringner et al., 2011), was used to validate the mRNA expression levels of four immunosuppressive factor genes identified from the screened cell lines of GSE40057.

## UALCAN and Kaplan–Meier survival analyses

UALCAN (http://ualcan.path.uab.edu/index.html), an interactive web resource that contains a large amount of cancer transcriptome data derived from TCGA and MET500 transcriptome sequencing (Chandrashekar et al., 2017), was used to explore the mRNA expression levels of 6 immunosuppressive genes identified from the screened GSE1561 tissue samples. Kaplan–Meier Plotter (http://kmplot.com/analysis/), an open web-based resource (Győrffy et al., 2013), was used to determine the relationship between the overall survival rate and mRNA expression levels of 6 immunosuppressive genes identified from the screened GSE1561 tissue samples.

## Immunohistochemistry

Vascular endothelial growth factor A (VEGFA) and MIF levels in normal and breast cancer tissues were evaluated by immunohistochemistry (IHC) using polyclonal antibodies (1:250 dilution, DF7470 and DF6404, Affinity Biosciences, Cincinnati, OH, USA) on commercial tissue arrays (Shanghai Outdo Biotech Co., Shanghai, China). The array consists of 10 normal and 90 breast tumor specimens. Each sample was given a modified histochemical score (MH score), which was assessed by both the proportion and the intensity of cells stained at each intensity, to reflect the staining intensity. The intensity of each grade is the average MH score of all samples in that grade.

## Cell culture and total RNA isolation

The breast cancer cell lines MDA-MB-231 and T47D were cultured in DMEM supplemented with 10% fetal bovine serum (FBS). MCF7 cells were cultured in RPMI-1640 medium with 10% FBS, and BT549 cells were cultured in RPMI-1640 medium with 0.023 IU/ml insulin and 10% FBS. Total RNA was extracted using TRIzol® reagent (Invitrogen, Carlsbad, CA, USA).

### qRT-PCR analysis of mRNA expression levels of CD274 and IL8

Two micrograms of total RNA was reverse-transcribed using the RevertAid™ First-Strand cDNA Synthesis Kit (Thermo, Boston, MA, USA). SYBR® Premix Ex Taq™II (TaKaRa, Shiga, Japan) was used to conduct quantitative RT-PCR. The primer sequences used for RT-PCR were as follows: CD274-Forward: CGTTGTGCTTGAACCCTTGA, CD274-Reverse: ACACAAGGAGCTCTGTTGGA; IL8-Forward: GAGACAGCAGAGCACACAAG, IL8-Reverse: TTGGGGTGGAAAGGTTTGGA; β-actin-Forward: GAACGGTGAAGGTGACAG, and β-actin-Reverse: TAGAGAGAAGTGGGGTGG. Each sample was analyzed in triplicate. According to the manufacturer's suggested protocols, Applied Biosystems® 7500 Real-Time PCR Systems (Thermo, Boston, MA, USA) were used for the real-time PCRs. The ΔΔCt method was used to calculate the fold change in gene expression.
*Western blot analysis of the protein expression level of CD274 and IL8*

Total proteins were extracted from cells in RIPA Lysis Buffer (Vazyme, Piscataway, NJ, USA) containing protease inhibitors. A total of 40 μg of protein from each sample was denatured, fractionated by 10% SDS-PAGE, and transferred to a PVDF membrane (Immobilon®-P Transfer Membrane, Millipore, Milan, Italy). After blocking nonspecific antigens with 5% skim milk solution, blots were incubated overnight at 4 °C with primary rabbit monoclonal antibodies against *IL8* (1:1,000 working dilution, DF6998, Affinity Biosciences, Cincinnati, USA), *CD274* (1:1,000 working dilution, DF6526, Affinity Biosciences, Cincinnati, USA), or β-actin (1:1,000 working dilution, Santa Cruz Biotechnology, Inc., Santa Cruz, CA, USA) in 5% skim milk 0.05% TBS-Tween 20 buffer. Antibody binding to the membrane was detected with a secondary antibody (goat anti-rabbit IgG 1:5,000, ZSGB Biosciences, Beijing, China) conjugated to horseradish peroxidase and visualized by enzyme-linked chemiluminescence (EasySee® Western Blot Kit, TransGen Biotechnology, Beijing, China) with the Scientific MYECL Imager (Thermo, Boston, MA, USA). Densitometric analysis performed with ImageJ software was used to normalize the signals of *IL8* and *CD274*. The intensity of the two bands was normalized against the signal of β-actin.

## Statistical analysis

Each experiment was repeated at least three times. DEGs were identified by the Bioconductor "limma" package using moderated *t*-tests ($p < 0.05$) and comparisons of multiple change (basal vs. luminal, GSE40057 > 2-fold, GSE1561 > 1.5-fold). The association of MIF and VEGFA expression and clinicopathological data were analyzed by one-way ANOVA in SPSS 17.0 (SPSS Inc. Chicago, IL, USA).

# RESULTS

## Identification of DEGs

To study the gene expression profiles of different breast cancer cell lines and tissue-based microarray datasets, GSE40057 and GSE1561 data from the GEO database were downloaded, re-modeled, analyzed and compared. According to the results of the PCA (Fig. 1A), the samples were successfully separated into three major groups: luminal-like (luminal A, luminal B), basal-like and edged samples. In the hierarchical clustering (Fig. 1B), eight cell lines (four basal-like and four luminal-like) from GSE40057 and 32 tissue samples (16 basal-like and 16 luminal-like) from GSE1561 were chosen based on the proximities between the samples for subsequent analyses. The results showed that 2,188 and 1,963 genes were significantly differentially expressed between the luminal-like and basal-like groups of cell lines and tissue samples, respectively (Table S1).

## WGCNA, GO, and KEGG pathway enrichment analyses of each gene module

To investigate what causes the difference in the degree of malignancy between basal-like and luminal-like breast cancer, the sets of genes related to basal- and luminal-like breast cancer were first screened by constructing a gene co-expression network by WGCNA.

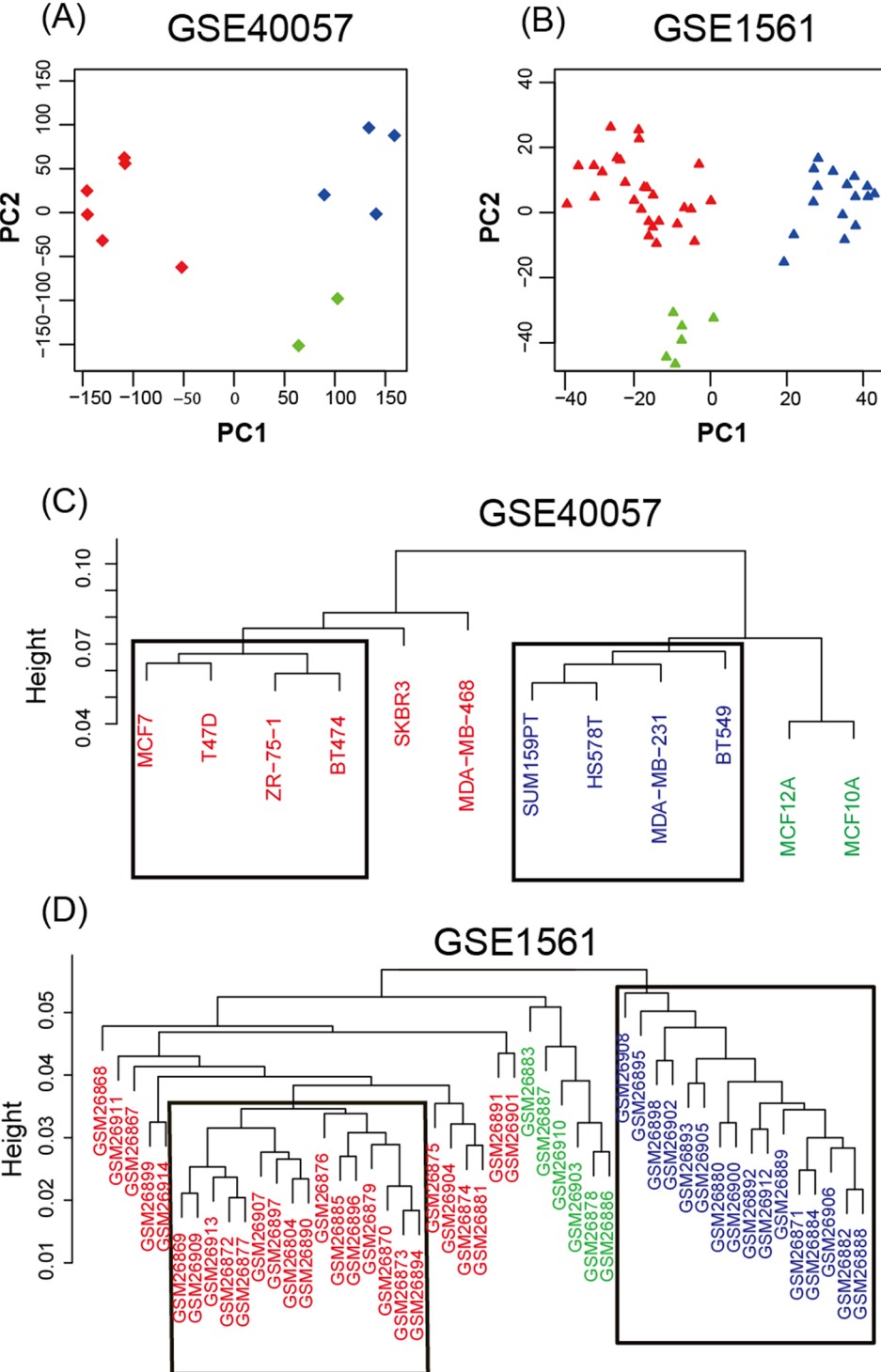

**Figure 1 Unsupervised analysis.** (A, B) Principal components of all genes. The first two PCAs are plotted. The three major groups are colored blue (basal-like), red (luminal-like), and green (edged samples). (C, D) Hierarchical clustering of all samples. According to the proximities between the samples, eight cell lines (four basal-like and four luminal-like) from GSE40057, and 32 tissue samples (16 basal-like and 16 luminal-like) from GSE1561 were chosen for subsequent analyses.

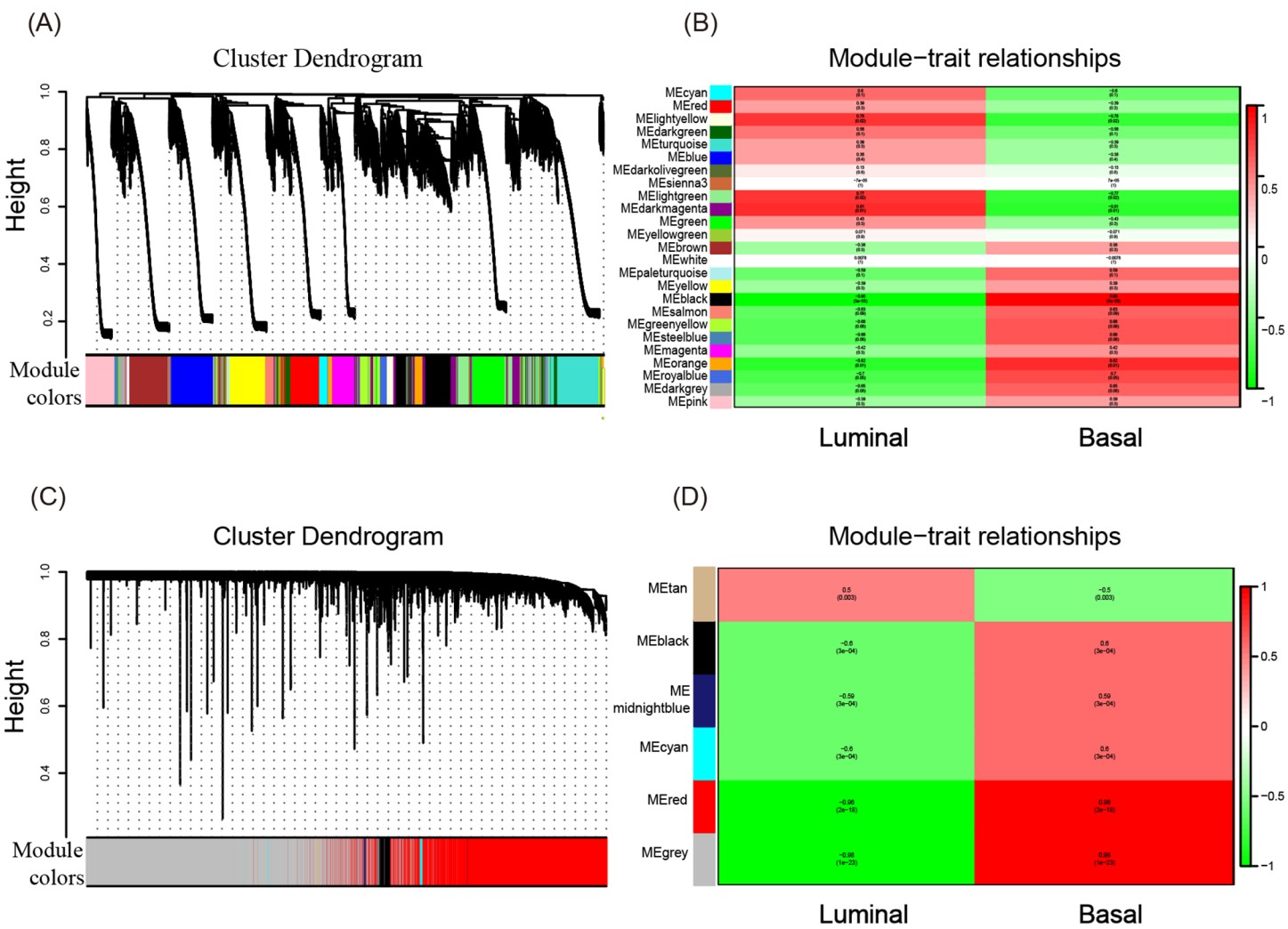

**Figure 2 WGCNA of GSE40057 (A, B) and GSE1561 (C, D).** (A) A total of 20,283 genes were assigned to 25 modules. The gene dendrogram is shown in the top portion, and the 25 gene modules are shown in the bottom portion. (B, D) The relationship between modules and traits. The upper score in each box represents the module significance (MS) score, and the lower value represents the corresponding *p*-value. If a box is red (MS > 0), the module in which the box is located is thought to be correlated with the trait; the lower the *p*-value, the higher the MS score is, and the redder the color of the box, the stronger the correlation is. (C) A total of 12,752 genes were assigned to six modules.     

Then, 25 and six gene modules for GSE40057 (cell lines) and GSE1561 (tissue samples) were identified, respectively (Figs. 2A and 2C). The member genes involved in the same module were highly interconnected and further analyzed in the GO and KEGG pathway enrichment analysis. For cell lines, the light yellow, light green, and dark magenta modules had significant correlations with luminal-like traits ($p < 0.05$); the black and orange modules had significant correlations with basal-like traits ($p < 0.05$) (Fig. 2B). For the tissue samples, the tan module had a significant correlation with luminal-like traits ($p < 0.05$), and the black, midnight blue, cyan, red, and gray modules had a significant correlation with basal-like traits ($p < 0.05$) (Fig. 2D). Interestingly, the subsequent enrichment analyses for the genes in each module showed that the enrichment results of

**Table 1 TOP 10 GO terms of GO functional annotations for genes in module black.**

| GO term_MF | Count | % | p-value |
|---|---|---|---|
| GO:0006955~immune response | 72 | 24.91 | 5.94E-53 |
| GO:0002250~adaptive immune response | 30 | 10.38 | 1.38E-23 |
| GO:0050776~regulation of immune response | 31 | 10.72 | 2.30E-22 |
| GO:0060333~interferon-gamma-mediated signaling pathway | 20 | 6.92 | 1.40E-18 |
| GO:0006954~inflammatory response | 37 | 12.80 | 4.16E-18 |
| GO:0050852~T cell receptor signaling pathway | 23 | 7.96 | 1.79E-15 |
| GO:0045087~innate immune response | 34 | 11.76 | 6.47E-14 |
| GO:0050900~leukocyte migration | 20 | 6.92 | 6.71E-14 |
| GO:0031295~T cell costimulation | 17 | 5.88 | 7.59E-14 |
| GO:0019882~antigen processing and presentation | 14 | 4.84 | 2.48E-12 |
| **GO term_BP** | | | |
| GO:0042605~peptide antigen binding | 11 | 3.81 | 6.26E-12 |
| GO:0004888~transmembrane signaling receptor activity | 18 | 6.23 | 3.51E-08 |
| GO:0004872~receptor activity | 18 | 6.23 | 4.30E-08 |
| GO:0008009~chemokine activity | 10 | 3.46 | 4.93E-08 |
| GO:0032395~MHC class II receptor activity | 7 | 2.42 | 5.49E-08 |
| GO:0005102~receptor binding | 22 | 7.61 | 1.30E-07 |
| GO:0042288~MHC class I protein binding | 7 | 2.42 | 2.83E-07 |
| GO:0019864~IgG binding | 6 | 2.08 | 3.50E-07 |
| GO:0023026~MHC class II protein complex binding | 6 | 2.08 | 3.11E-06 |
| GO:0003823~antigen binding | 11 | 3.81 | 4.10E-06 |

**Note:**
  GO, gene ontology; BP, biological process; MF, molecular function.

**Table 2 TOP 10 clusters of KEGG pathway enrichment analysis for genes in module black.**

| KEGG_pathway | Count | % | p-value |
|---|---|---|---|
| hsa05330:Allograft rejection | 19 | 6.57 | 9.37E-20 |
| hsa04612:Antigen processing and presentation | 24 | 8.30 | 2.51E-19 |
| hsa05332:Graft-vs.-host disease | 18 | 6.23 | 2.87E-19 |
| hsa05416:Viral myocarditis | 20 | 6.92 | 5.00E-17 |
| hsa04940:Type I diabetes mellitus | 18 | 6.23 | 5.08E-17 |
| hsa05150:Staphylococcus aureus infection | 19 | 6.57 | 3.46E-16 |
| hsa05320:Autoimmune thyroid disease | 18 | 6.23 | 3.69E-15 |
| hsa05152:Tuberculosis | 27 | 9.34 | 2.08E-13 |
| hsa04145:Phagosome | 25 | 8.65 | 2.78E-13 |
| hsa04514:Cell adhesion molecules (CAMs) | 24 | 8.30 | 6.94E-13 |

the black module in the tissue samples were relatively uniform in immune-related events, such as T cell co-stimulation, peptide antigen binding, leukocyte migration (Table 1) and allograft rejection, antigen processing and presentation, and graft-vs.-host disease (Table 2). Genes in other modules did not show the same uniform enrichment results in immune-related or any other specific aspect.

## The expression of immune-related genes and immune-oncology targets

According to the cancer inflammation & immunity crosstalk PCR array profile from Qiagen (https://www.qiagen.com/us/products/discovery-and-translational-research/pcr-qpcr/qpcr-assays-and-instruments/mrna-incrna-qpcr-assays-panels/rt2-profiler-pcr-arrays/?catno=PAHS-181Z#geneglobe), a total of 85 key genes (Table S2), including 16 immunosuppressive factors, were enrolled for subsequent analysis. The Venn diagram (Fig. 3A) shows the intersection between immune-related genes and DEGs from the GSE40057 and GSE1561 datasets. The Venn diagram (Fig. 3B) shows the intersection between 29 clinical IO targets (Table S2) (Szekely et al., 2018) and DEGs from GSE40057 and GSE1561. There were 15 immune-related genes in GSE40057 (Table S3), including four immunosuppressive factors, *CD274* (*PDL1*), *CSF2*, *IL8* (*CXCL8*), and *TGFβ1*; 31 immune-related genes in GSE1561 (Table S3), including six immunosuppressive factors, *CXCL12*, *CXCL5*, *IDO1*, *MIF*, *PTGS2*, and *VEGFA*; and six immune-related genes in both; two IO targets in GSE40057, *CD274* and *IL8*; and two IO targets in GSE1561, *CXCL12*, and *IDO1*. Interestingly, compared with the luminal-like cell lines and tissue samples, most immune-related genes identified in the basal-like malignancies were upregulated, except *CXCL12* (Figs. 3C and 3D).

## GOBO analysis for the immunosuppressive factors screened from cell lines

GOBO analysis showed that *CSF2*, *IL8*, and *TGFβ1* expression levels were inconsistent across the cell lines; there was no information in GOBO for *CD274*. *IL8* had higher average expression levels in basal-like cell lines ($p < 0.01$) (Fig. 4A); *TGFβ1* had higher average expression levels in basal-like ($p < 0.01$) (Fig. 4A) and triple-negative cell lines ($p < 0.01$) (Fig. 4B). The mRNA levels of *CSF2, IL8*, and *TGFβ1* across the 51 breast cancer cell lines are also shown (Fig. 4C).

## UALCAN and Kaplan–Meier survival analyses for the immunosuppressive factors identified from the screened tissue

The UALCAN analysis showed that only *CXCL12* (Fig. 5B) had a lower expression level in basal-like (triple-negative) breast cancer compared with luminal-like breast cancer, while *CXCL5* (Fig. 5A), *IDO1* (Fig. 5C), *MIF* (Fig. 5D), *PTGS2* (Fig. 5E), and *VEGFA* (Fig. 5F) all had increased expression levels in basal-like breast cancer.

The analysis of overall survival showed that higher *CXCL12* (Fig. 6B), *IDO1* (Fig. 6C), and *PTGS2* (Fig. 6E) mRNA expression levels correlated with a comparatively higher survival rate ($p < 0.05$), while higher *MIF* and *VEGFA* (Figs. 6D and 6F) expression levels correlated with a lower survival rate ($p < 0.01$). A further analysis of the screened immunosuppressive factors showed that the combination of *CXCL12*, *IDO1*, and *PTGS2* was correlated with an increased survival rate (weight: 1:1:1; $p < 0.01$; Fig. 6G), while the combination of *MIF* and *VEGFA* was correlated with a reduced survival rate (weight: 1:1; $p < 0.01$; Fig. 6H).

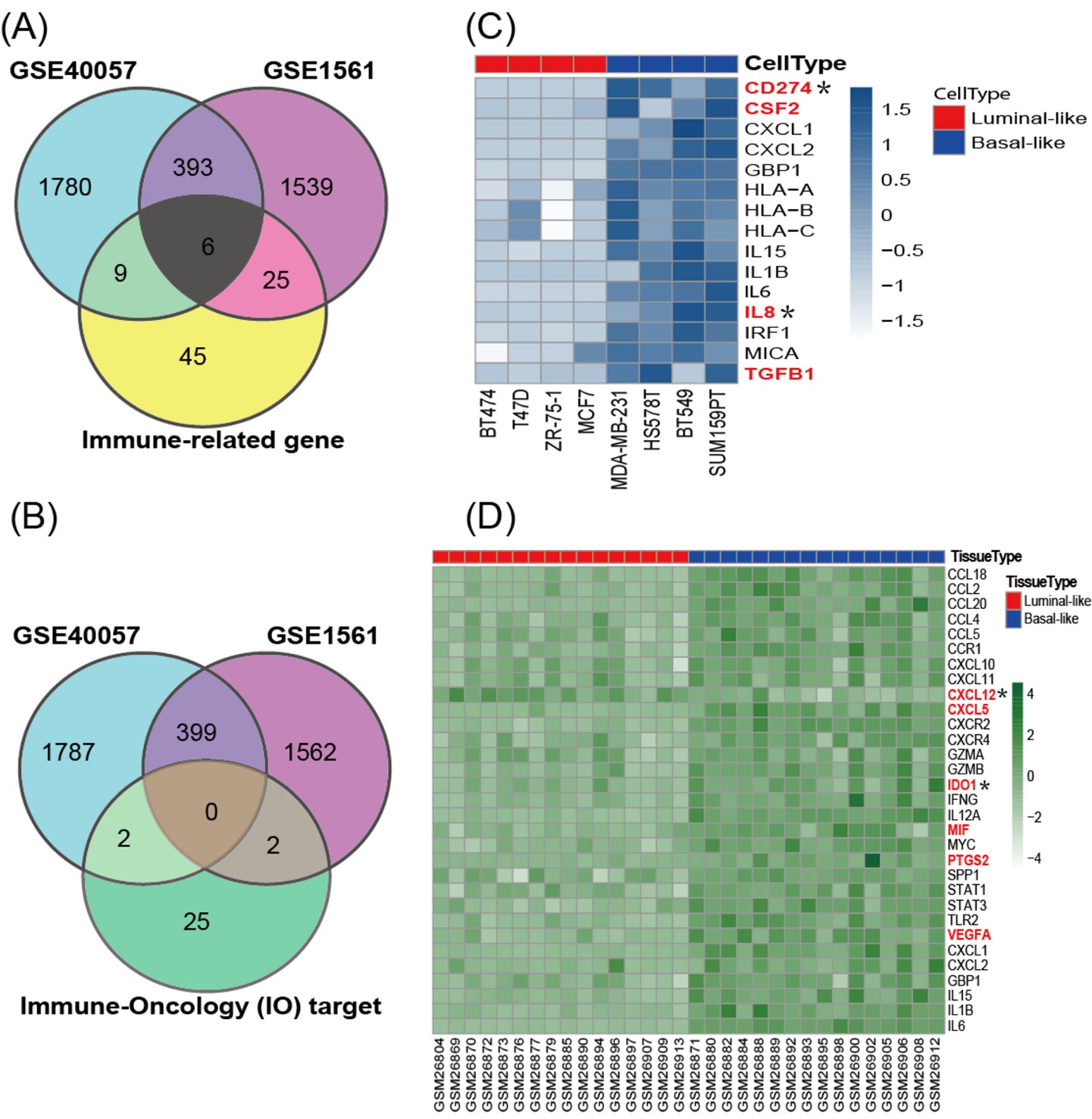

**Figure 3 The expression of immune-related genes in cell lines and tissue samples.** (A) The Venn diagram shows the intersection between immune-related genes and DEGs from GSE40057 and GSE1561. (B) The Venn diagram shows the intersection between 29 clinical IO therapeutic targets and DEGs from GSE40057 and GSE1561: CD274 and IL8 from GSE40047; CXCL12 and IDO1 from GSE1561. (C, D) Hierarchical clustering analysis of immune-related gene expression in cell lines (C) and tissue samples (D). Each row corresponds to an immune-related gene, and each column corresponds to an independent cell or tissue sample. The darker color indicates increased expression, and the immune-related genes in red letters are thought to be immunosuppressive factors. *IO target.

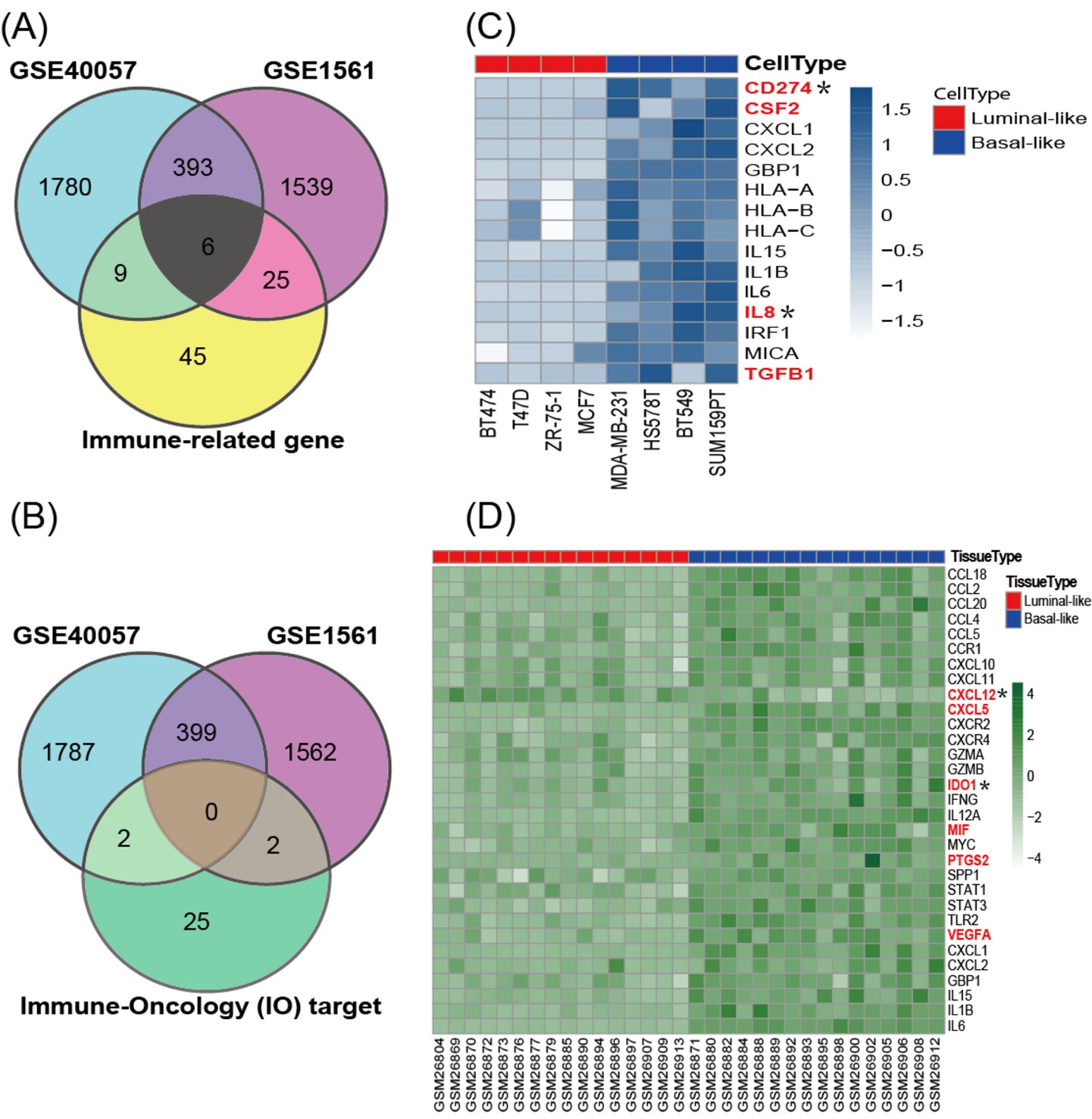

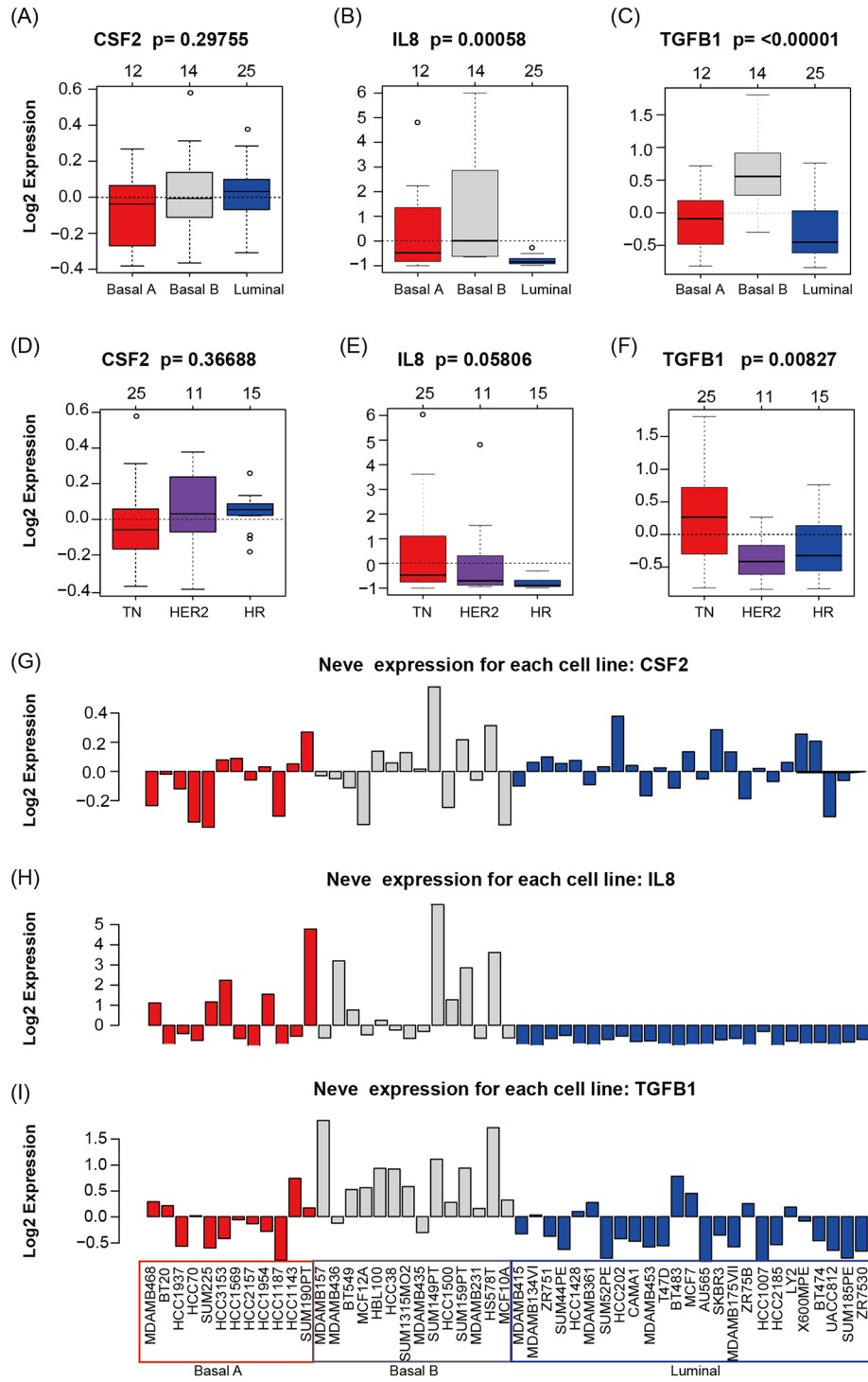

**Figure 4 CSF2, IL8, and TGFβ1 expression in human breast cancer cell lines with GOBO analysis.** (A–C) Box plots of CSF2, IL8, and TGFβ1 expression across 51 breast cancer cell lines grouped into basal A (red), basal B (gray), and luminal (blue) subgroups. The expression levels of IL8 and TGFβ1 in the basal A and B subgroups were higher than those in the luminal subgroup ($p < 0.01$), while the CSF2 results were not statistically significant. (D–F) Box plots of CSF2, IL8, and TGFβ1 expression across 51 breast cancer cell lines grouped into triple-negative (TN), HER2-positive and hormone receptor-positive (HR) groups. The expression level of TGFβ1 in the TN subgroup was higher than that in the HER2 and HR subgroups ($p < 0.01$), while the CSF2 and IL8 results were not statistically significant. (G–I) CSF2, IL8, and TGFβ1 mRNA levels across 51 breast cancer cell lines.

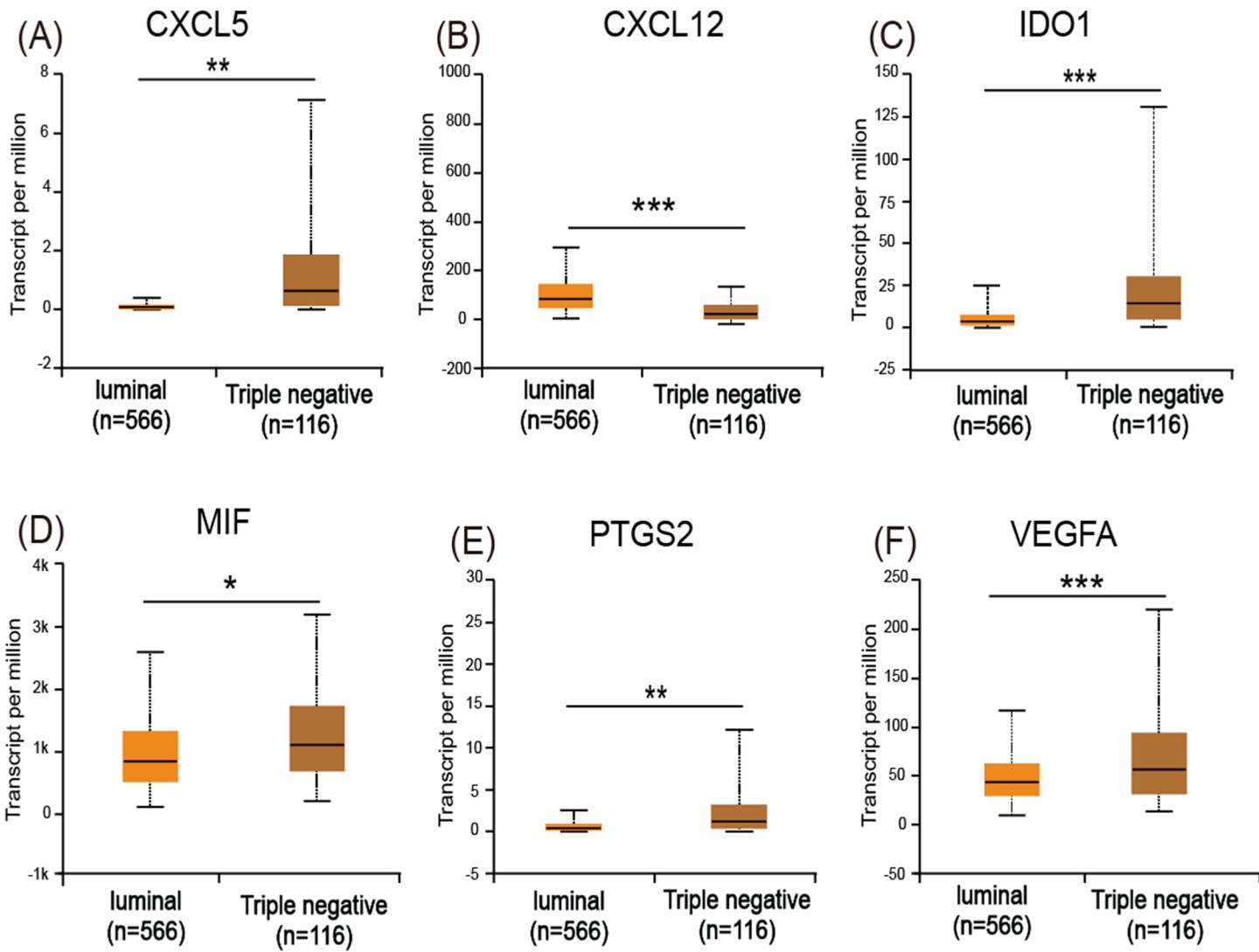

**Figure 5 Expression of the six immuosuppressive factors in breast cancer based on breast cancer subtypes.** (A, C, D, E, and F) CXCL5, IDO1, MIF, PTGS2, and VEGFA are expressed at higher levels in basal (triple-negative) breast cancer compared with luminal breast cancer. (B) CXCL12 is expressed at lower levels in basal (triple-negative) breast cancer. $*p < 0.05$, $**p < 0.005$, and $***p < 0.001$.

### Expression of MIF and VEGFA in breast tissue microarrays

To validate whether MIF and VEGFA protein expression is associated with breast cancer malignancy, immunohistochemical detection was performed in a tissue microarray with 90 primary tumor tissues and 10 normal breast tissue samples ($p < 0.01$; Fig. 7). The results showed that MIF expression was increased dramatically in the metastasis group ($p < 0.05$) and VEGFA expression positively correlated with tumor grade ($p < 0.05$; Table 3).

### CD274 (PD-L1) and IL8 are highly expressed in basal-like breast cancer cell lines

To validate the expression pattern of the two IO targets, *CD274* and *IL8*, qRT-PCR, and western blotting were used to detect the expression of mRNA and protein from two

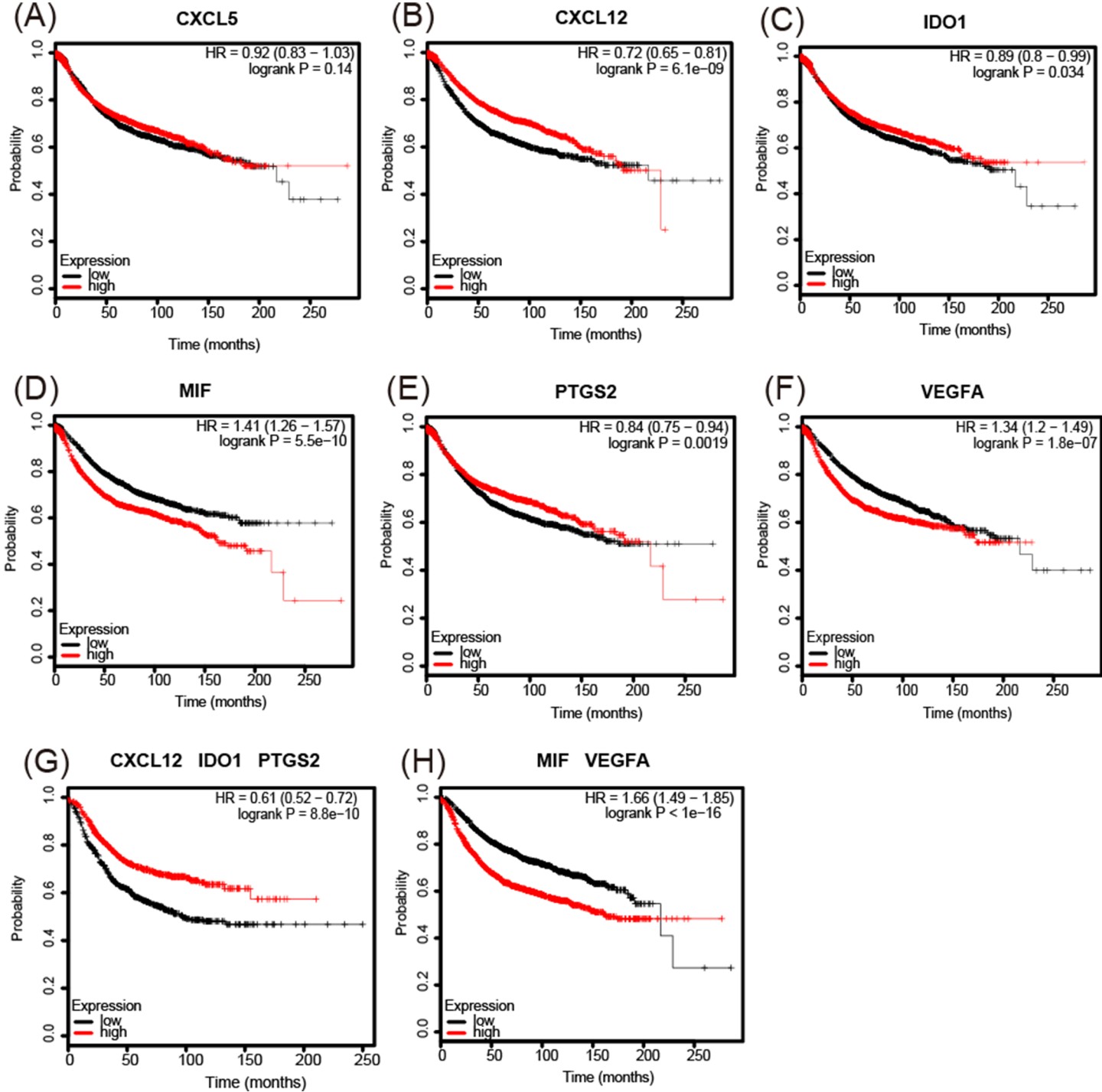

**Figure 6 Kaplan-Meier Plotter determined the relationship between the overall survival rate and mRNA expression levels of 6 immunosuppressive factors using microarray data from 3,951 patients.** (A) There was no significant correlation between CXCL5 expression and survival rate. (B–F) Increased CXCL12, IDO1, and PTGS2 mRNA expression levels correlated with a comparatively increased survival rate ($p < 0.05$), while increased MIF and VEGFA expression levels correlated with a reduced survival rate ($p < 0.01$). (G, H) The combination of CXCL12, IDO1, and PTGS2 correlated with an increased survival rate (weight: 1:1:1) ($p < 0.01$), while the combination of MIF and VEGFA correlated with a reduced survival rate (weight: 1:1) ($p < 0.01$). HR = hazard ratio.

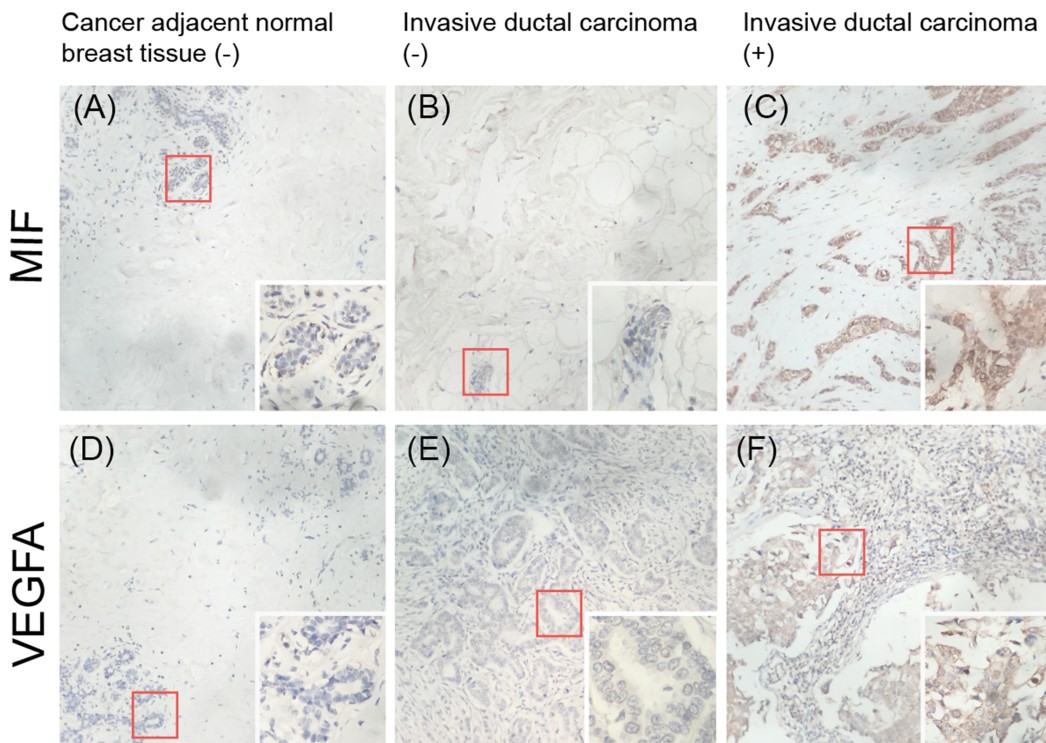

**Figure 7 Immunohistochemical detection of the expression of MIF and VEGFA in a breast cancer tissue microarray.** (A, D) Negative expression (−) in cancer-adjacent normal breast tissue. (B, E) Negative expression in invasive ductal carcinoma. (C, F) Positive expression (+) in invasive ductal carcinoma (original magnification ×200; inset ×400).

**Table 3 Relationship between MIF, VEGFA expression level, and clinico-pathologic parameters of breast cancer by tissue microarray.**

| Variable | Number of cases | MIF, 100% | | P | VEGFA, 100% | | P |
|---|---|---|---|---|---|---|---|
| | | High | Low | | High | Low | |
| **Pathologic grade** | | | | | | | |
| 1 | 16 | 11 (68.8) | 5 (31.2) | 0.397 | 6 (37.5) | 10 (62.5) | 0.017* |
| 2,3 | 58 | 32 (55.2) | 26 (44.8) | | 42 (72.4) | 16 (27.6) | |
| **Clinical stage** | | | | | | | |
| I | 17 | 3 (17.6) | 14 (82.4) | 0.095 | 12 (70.6) | 5 (29.4) | 1.000 |
| II, III | 73 | 30 (41.1) | 43 (58.9) | | 51 (69.9) | 22 (30.1) | |
| **Lymph node status** | | | | | | | |
| No metastasis | 78 | 25 (32.1) | 53 (67.9) | 0.027* | 48 (61.5) | 30 (38.5) | 0.524 |
| Metastasis | 12 | 8 (66.7) | 4 (33.3) | | 9 (75.0) | 3 (25.0) | |

**Notes:**
The total number of samples in pathologic grade does not equal 90, as some samples are not included in any given grades.
* $p < 0.05$.

basal-like (BT549 and MDA-MB-231) and two luminal-like (MCF7 and T47D) breast cancer cell lines (*Neve et al., 2006*). The qRT-PCR results (Fig. 8A) showed that *CD274* and *IL8* were upregulated in the basal-like breast cancer cell lines BT549 and MDA-MB-231

($p < 0.01$). Similar to the qRT-PCR results, the western blot analysis (Fig. 8B) indicated that CD274 and IL8 protein levels were increased in the BT549 and MDA-MB-231 cell lines compared to those in the MCF7 and T47D cell lines.

## DISCUSSION

The breast tumor microenvironment consists of epithelial tumor cells and ECM, including stromal cells such as fibroblasts, adipocytes, endothelial cells and resident immune cells; a multitude of soluble factors; and more recently identified regulatory mediators, such as microRNAs, metabolites, and exosomes (*Dittmer & Leyh, 2015*). Although cancer progression has been associated with genetic mutations and epigenetic changes in tumor cells, increasing evidence suggests that it is not entirely driven by cancer cell processes and may be influenced by the interplay between cancer cells and their surrounding microenvironment (i.e., tumor-stroma crosstalk) (*Criscitiello, Esposito & Curigliano, 2014*). There is evidence demonstrating that both the stroma and tumor cells evolve upon tumor initiation and progression, which makes the tumor-stroma environment distinct from that of healthy tissue (*Quail & Joyce, 2013*). Upon its conversion from normal stroma, tumor stroma thwarts anticancer activities, and promotes cancer progression (*Granot & Fridlender, 2015*). Therefore, the molecular changes in tumor cells often do not reflect all the changes that occur during tumor-stroma crosstalk in the microenvironment (*Morandi & Chiarugi, 2014*).

In this paper, two original datasets, GSE40057 and GSE1561, were downloaded from the GEO database. To avoid the inaccuracy produced by "edged" samples, only eight cell lines from GSE40057 and 32 tissue samples from GSE1561, representing basal-like and luminal-like groups, were chosen for subsequent analyses due to their great difference in malignancy. In the WGCNA, all genes of the microarray were assigned to corresponding modules based on the weighted gene co-expression network, and the correlations of each module with luminal and basal traits were calculated. Genes involved in the same module were found to be highly interconnected and relate to a specific trait, which is different from DEGs that only showed differences in expression levels between groups. Therefore, we chose the WGCNA instead of conventional differential gene expression analysis to screen biological characteristics related to the malignant phenotype of breast cancer. GO and KEGG pathway enrichment analyses for each module showed that genes in the black module of the tissue samples were uniformly enriched in immune-related events, and this module showed a closer correlation with basal-like rather luminal-like breast cancer, indicating that there are certain immune-related events that may play pivotal roles in the malignant phenotype of basal-like breast cancer. However, the enrichment results of modules in the cell lines were not clearly related to immune-related events. This may be because of the tumor-stroma crosstalk in vivo, which is not represented in the cell lines. Therefore, in the field of tumor immune research, the selection of model cell lines or tissue samples may lead to different results, and both have their own unique advantages in scientific research. For instance, the immunosuppressive factors *CD274* and IL8 identified from the cell lines were also validated by UALCAN and

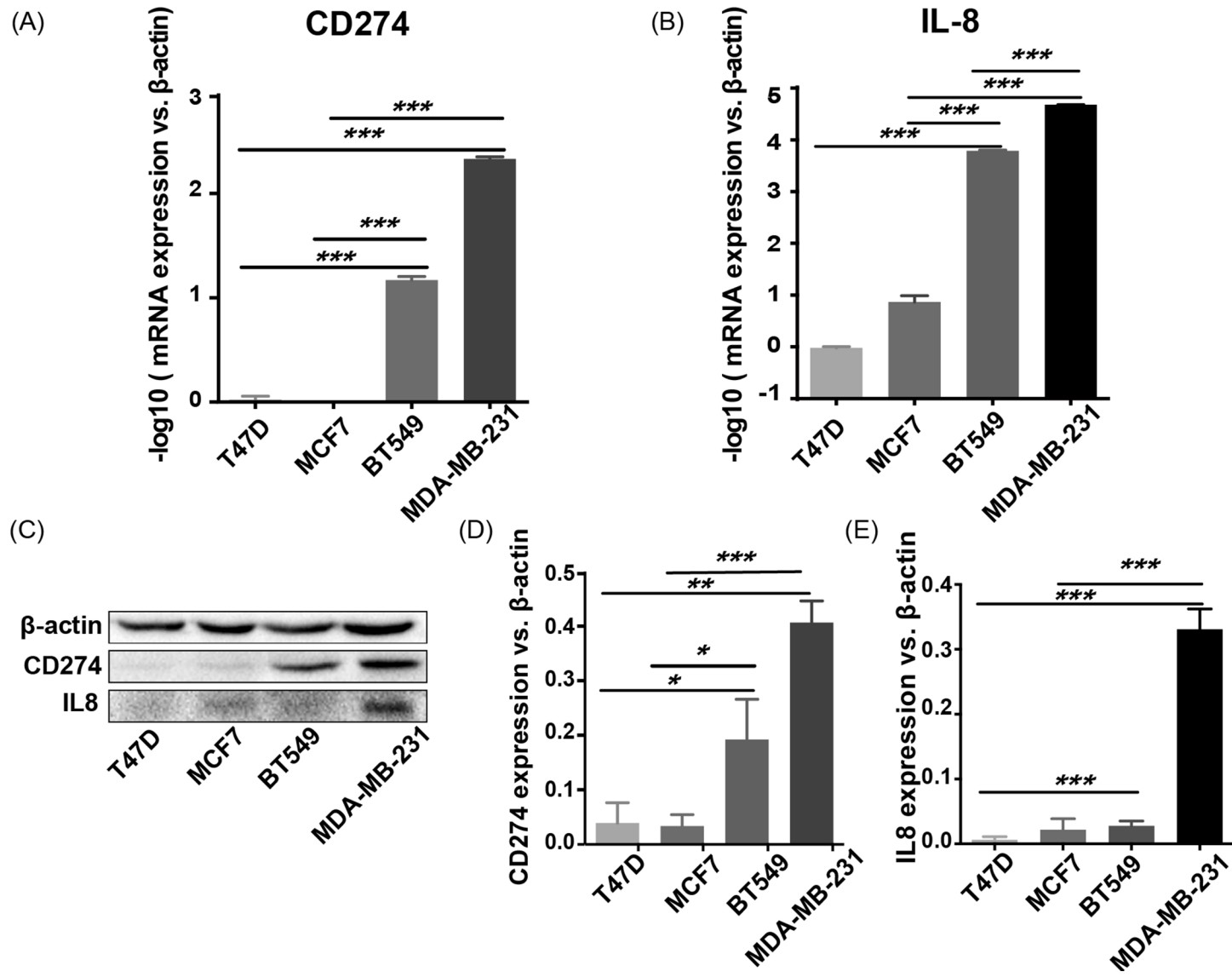

**Figure 8 qRT-PCR and western blot results.** (A, B) qRT-PCR results showed that CD274 and IL8 were upregulated in the basal-like breast cancer cell lines BT549 and MDA-MB-231 ($p < 0.0001$). Similar to the qRT-PCR results, the western blot analysis (C–E) indicated that CD274 and IL8 protein were increased in the BT549 and MDA-MB-231 cell lines compared to the MCF7 and T47D cell lines. ***$P < 0.001$, **$p < 0.01$, and *$p < 0.05$.

Kaplan–Meier Plotter, which contained data from tumor tissue rather than the cell lines. This result shows that *CD274* and *IL8* are also highly expressed in basal-like tumors; increased IL8 expression is associated with poor prognosis, while opposing associations are observed for *CD274* (Fig. S1). The result of the *CD274* analysis did not meet the expectation that higher *CD274* expression is associated with poor prognosis. This may be because the tumor tissues contain a variety of stromal cells, such as fibroblasts, adipocytes, endothelial cells and resident immune cells, and *CD274* is mainly expressed by antigen-presenting cells, which need to interact with their receptors in the surrounding

stromal cells to exert their effects (*Freeman et al., 2000*). Therefore, when we use these databases to validate the results from cell lines, which are tumor cells, there may be some inconsistencies. This is also the reason why we analyzed and validated cell lines (pure) and tumor tissues (mixed), separately.

Tumor cells have many strategies for avoiding immune attack, including decreased tumor antigen expression by HLA molecules on the tumor cell surface, downregulation of tumor antigen presentation by dendritic cells, the release of immunosuppressive factors and the activation of regulatory T cells (*Raposo et al., 2015*). Among the many evasion strategies, the release of immunosuppressive factors to induce immunosuppression is an important mechanism for tumor cell evasion of immune surveillance (*Jiang & Shapiro, 2014*). Immunosuppression is a reduction of the activation or efficacy of the immune system (*Schlößer et al., 2014*). Both tumor cells and stromal cells can be induced to synthesize and/or secrete immunosuppressive factors to evade immune surveillance, contributing to tumor initiation and progression (*Grivennikov, Greten & Karin, 2010*). To date, more than 20 immunosuppressive factors produced by tumor and/or stromal cells have been discovered, including transforming growth factor-β1 (TGFβ1) (*Wang et al., 2015*), prostaglandin E2 (*Kalinski, 2012*), vascular endothelial growth factor (VEGF) (*Shibuya, 2013*), interleukin-10 (*Geginat et al., 2016*), interleukin-4 (*Egawa et al., 2013*), cyclooxygenase-2 (*Li et al., 2013*), programmed cell death 1 (*Gatalica et al., 2014*), and cytotoxic T-lymphocyte associated antigen 4 (*Lan et al., 2013*). Although therapeutic agents that target immunosuppressive factors, such as BMS-936559, pidilizumab and ipilimumab, have achieved breakthrough responses in cancer immunotherapy and represent one of the most promising strategies for tumor treatment (*Schlößer et al., 2014*), other immunosuppressive targets are still under investigation, as their roles in tumor malignancy are not completely understood.

The analysis of cell lines showed that *CD274*, *CSF2*, *IL8*, and *TGFβ1* were upregulated in basal-like cell lines compared with luminal-like cell lines. Furthermore, GOBO analysis confirmed that *IL8* and *TGFβ1* had higher average expression levels in basal-like and triple-negative cell lines, indicating that *IL8* and *TGFβ1* may be associated with a malignant phenotype. The analysis of tissue samples revealed that *CXCL5*, *IDO1*, *PTGS2*, *MIF*, and *VEGFA* were upregulated in basal-like tissues compared with luminal-like tissues, which was also confirmed by UALCAN analysis. In addition, the Kaplan–Meier survival analysis showed that increased *MIF* and *VEGFA* expression levels in breast cancer patients correlated with a reduced survival rate. These findings suggest that the increased expression levels of *MIF* and *VEGFA* contribute to the malignant phenotype of breast cancer. Our IHC results also validated this result.

Cancer-derived IL8 may result in the recruitment and activation of tumor-associated neutrophils and myeloid-derived suppressor cells to contribute to the tumor microenvironment and immune suppression and activate endothelial cells for angiogenesis (*Waugh & Wilson, 2008*). IL8 functions by activating the PI3K-Akt and PLC-PKC signaling pathways. These two signaling pathways have been demonstrated to be associated with angiogenesis, cell survival, and migration (*Cheng et al., 2008*). Overexpressed IL8 is associated with accelerated breast cancer progression, an increased

tumor load, and the presence of distant metastasis, ultimately leading to poor survival (*Singh et al., 2013*). TGFβ1 is a well-known family involved in various tumor processes, such as induction of epithelial–mesenchymal transition and the regulation of cancer migration and invasion (*Xu, Lamouille & Derynck, 2009*). The overexpression of MIF has been demonstrated in the progression of multiple cancers, such as ovarian cancer (*Krockenberger et al., 2012*), hepatocellular carcinoma (*Wang et al., 2014*), gastric cancer (*He et al., 2006*), and other malignant cancers. Several studies have elucidated the multiple roles of MIF in the breast cancer microenvironment, including increasing the recruitment of immunosuppressive cells (*Simpson, Templeton & Cross, 2012*), inducing angiogenesis and breast cancer cell trans-endothelial migration (*Martinez et al., 2014*). In addition, MIF also acts on tumor cells to facilitate cell proliferation and cell survival (*Lue et al., 2007*). Thus far, the VEGFA protein has been identified as a major factor that contributes to tumor angiogenesis and malignant progression in a variety of cancers (*Li et al., 2017*; *Yang et al., 2018*). In recent years, a number of antiangiogenic drugs have been designed and showed significant effects in chemotherapy. Unfortunately, some of these drugs (e.g., bevacizumab) showed limited effects in specific breast cancer conditions (*Bergh et al., 2012*; *Varinska et al., 2015*). Therefore, patients need to be evaluated before the implementation of antiangiogenic therapy. Prospectively, VEGFA may act as an evaluation factor to identify breast cancer patients who might benefit from antiangiogenic therapy.

Currently, a variety of IO targets are available in clinical therapy (*Szekely et al., 2018*). Among the genes identified in our results, *CD274, IL8, CXCL12*, and *IDO1* (Fig. 3B) are included in these IO targets. Since the Kaplan–Meier survival analysis showed that CXCL12 and IDO1 were not associated with the malignant phenotype of breast cancer, they were not studied further. The high expression levels of *CD274* and *IL8* in basal-like breast cancer were validated by qRT-PCR and western blot, suggesting that clinical *CD274*- and *IL8*-targeting therapies may be more suitable for basal-like breast cancer. Other immunosuppressive factors in this study are also worth further study in the field of subtype-specific IO targeted therapy for breast cancer.

## CONCLUSIONS

Using online databases, model reconstruction and comparisons of the mRNA expression profiles of luminal-like and basal-like cell lines and primary breast cancer tissues, four immunosuppressive factors associated with a malignant phenotype in breast cancer were identified and validated. Such molecules could be used as biomarkers for malignant breast cancer phenotypes and prognosis. In addition, two immunosuppressive factors were confirmed as clinical IO therapeutic targets, which may be most suitable for the treatment of basal-like breast cancer. However, because the majority of immune-related factors have diverse roles in disease pathology and we still lack a complete understanding of the relationship between immunosuppressive factors and breast cancer malignancy, the feasibility of the clinical application of the identified factors as drug targets and prognosis predictors warrants further investigation.

## ACKNOWLEDGEMENTS

The authors would like to thank Ping Zhang, Qingmei Zhong, Xianhe Yang, Wu Wang, Di Yao, and Yingqun Xiao at Department of Pathology, Affiliated Infectious Diseases Hospital, Nanchang University, for their technical assistance.

### Funding

This work was supported by National Natural Science Foundation of China (No. 81160248, 81360313, 81560464) (to Daya Luo and Zhuoqi Liu), Natural Science Foundation of Jiangxi Province (No. 20151BAB205058, 20171BAB205055) (to Daya Luo and Zhuoqi Liu), Innovation Foundation for Graduate Students of Nanchang University (No. CX2015181) (to Yunlei Song). There was no additional external funding received for this study. The funders had no role in study design, data collection and analysis, decision to publish, or preparation of the manuscript.

### Grant Disclosures

The following grant information was disclosed by the authors:
National Natural Science Foundation of China: 81160248, 81360313, 81560464.
Natural Science Foundation of Jiangxi Province: 20151BAB205058, 20171BAB205055.
Innovation Foundation for Graduate Students of Nanchang University: CX2015181.

### Competing Interests

The authors declare that they have no competing interests.

### Author Contributions

- Ping Wang performed the experiments, analyzed the data, contributed reagents/materials/analysis tools, prepared figures and/or tables, authored or reviewed drafts of the paper, approved the final draft.
- Jiaxuan Liu approved the final draft, analyzed the data, prepared figures and/or tables.
- Yunlei Song performed the experiments, approved the final draft.
- Qiang Liu contributed reagents/materials/analysis tools, approved the final draft.
- Chao Wang approved the final draft.
- Caiyun Qian approved the final draft, giving suggestion.
- Shuhua Zhang approved the final draft, giving suggestion.
- Weifeng Zhu approved the final draft, giving suggestion.
- Xiaohong Yang approved the final draft, giving suggestion.
- Fusheng Wan conceived and designed the experiments, approved the final draft.
- Zhuoqi Liu performed the experiments, analyzed the data, contributed reagents/materials/analysis tools, prepared figures and/or tables, authored or reviewed drafts of the paper, approved the final draft.
- Daya Luo conceived and designed the experiments, authored or reviewed drafts of the paper, approved the final draft.

## Data Availability

The raw data is available at GenBank: GSE40057 and GSE1561.

## Supplemental Information

Supplemental information for this article can be found online at http://dx.doi.org/10.7717/peerj.7197#supplemental-information.

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
