# Peer review of "Screening of immunosuppressive factors for biomarkers of breast cancer malignancy phenotypes and subtype-specific targeted therapy"

_PeerJ, doi:10.7717/peerj.7197_

## Round 0.1 · original submission · Major Revisions

Dear Dr. Luo. Two of our reviewers have checked your manuscript and have pointed out a few major concerns that need to be addressed. Kindly address each point from the reviewers, including the explanations sought, missing labels, quantitations, and additional experiment suggested. Provide further explanations for the results as requested. Also please have the manuscript thoroughly examined for grammatical errors.

Reviewer 1 ·

Basic reporting

Good

Experimental design

Fine

Validity of the findings

Fine

Additional comments

The manuscript entitled “Screening of immunosuppressive factors for biomarkers of breast cancer malignant phenotypes and subtype-specific targeted therapy” is a well written manuscript. In this manuscript authors have tried to screen and validate immunosuppressive factors in luminal- and basal-like breast cancer cell lines and tissue samples associated with malignant phenotypes. I have following comments to improve the quality of this manuscript:

1. What would be the survival curve (overall survival ) if combination of the screened Immunosuppressive factors are plotted together for CXCL12 and CXCL5 (both Low vs high) or CXCL12 and IDO1, , CXCL5 and PTGS2 & VEGFA and MIF.
2. In your analysis, luminal-like breast cancer tissue contains both luminal A and B?
3. I would suggest quantifying staining (IHC) of MIF and VEGFA in Fig. 7 and plotting it side by side.

Minor changes:
1. Lane 43. The expression monitoring arrays raw data for were downloaded from Gene Expression... For must be removed.
2. Lane 237: stromal cells can be induced to synthesize and/or secret immunosuppressive factors to evade…. It should be secrete.

·

Basic reporting

'Screening of immunosuppressive factors for biomarkers of breast cancer malignant phenotypes and subtype-specific targeted therapy' by Liu et al., is mostly clearly written with relevant literature references. The author can make better use of figure legends to describe data.

Experimental design

Experiments are well designed, however authors should work on interpretation and presentation style.

Validity of the findings

It is an interesting manuscript, but I think authors have not done a great job of explaining their results and giving it larger clinical perspective. I have few comments which could be helpful to revise the current manuscript.
1) Figure-1: is not adequately discussed/explained in text/legends, it is not clear how authors determine basal-like (blue) and luminal-like (red) groups in tumor samples (GSE1561). PCA plot (1A) also shows four green color triangles (what is the identity of this group?) while hierarchal clustering shows many more unidentified patients (assuming they are green too). Authors should annotate all the patients (and cell lines) in Red, Green and Blue color to main the consistency between both panels.
2) Figure-2: It is not clear why there are more module in cell line data while less module in tumor data. Statement 'For cell lines, module lightyellow, lightgreen, darkmagenta etc.' is ambiguous. Which modules define the basal and luminal like behaviour should be stated clearly in text or figure legend.
3) Figure-3: It appears that author took lead from figure-2 where 'module black in tissue samples were relatively uniform in immune-related events' and investigated immune genes in cell line and tumor data. Since overlap with immune-related and IO gene is very modest author should apply statistical test to what is probability of finding such association by simple chance event. Author may like consider test genes associated with un-related models to test the significance of association with immune genes.
4) Figure-4: Author may like to validate some of the GOBO observations by qPCR analysis in cell lines. In text data should be discussed as described in each panel by specifying observations made in each panel (such as Fig.4A, etc).
5) Figure-5: Panel names are missing from figures, authors can should guide reader via panels, such as 'UALCAN analysis showed that only CXCL12 (Fig. 5B) has lower expression level in basal-like..'
6) Figure-6: Authors claim that 'Analysis of overall survival showed that higher CXCL12, CXCL5, IDO1 and PTGS2 mRNA expression levels were correlated with a comparatively higher survival rate (p<0.01),' However, Figure 6 (Since there are no panel names, I assume-A) CXCL5 P value is 0.14 which is not <0.01 please correct this.
7) Figure-7: Authors should also use markers to identify which cells are expressing VEGFA and MIF.
8) Figure-8: Is CD274 and IL-8 are also highly expressed in Basal-like tumors? Also does higher CD274 and IL-8 expression is associated with poor prognosis?

External reviews were received for this submission. These reviews were used by the Editor when they made their decision, and can be downloaded below.

---

## Round 0.2 · accepted · Accept

Our reviewers have gone through your rebuttal and modified manuscript and they have deemed it fit for acceptance. Congratulations.

Reviewer 1 ·

Basic reporting

Good

Experimental design

Good

Validity of the findings

N/A

Additional comments

Authors have incorporated all the suggested changes.